# The Multifunctional Nature of the MicroRNA/AKT3 Regulatory Axis in Human Cancers

**DOI:** 10.3390/cells12222594

**Published:** 2023-11-09

**Authors:** Chun Yang, Pierre Hardy

**Affiliations:** 1Research Center of CHU Sainte-Justine, University of Montréal, Montreal, QC H3T 1C5, Canada; cyang_09@yahoo.com; 2Department of Pharmacology and Physiology, Department of Pediatrics, University of Montréal, Montreal, QC H3T 1C5, Canada

**Keywords:** cancer, microRNA, AKT3, AKT signaling, small non-coding RNAs, circRNA, lncRNA, epigenetic regulation

## Abstract

Serine/threonine kinase (AKT) signaling regulates diverse cellular processes and is one of the most important aberrant cell survival mechanisms associated with tumorigenesis, metastasis, and chemoresistance. Targeting AKT has become an effective therapeutic strategy for the treatment of many cancers. AKT3 (PKBγ), the least studied isoform of the AKT family, has emerged as a major contributor to malignancy. AKT3 is frequently overexpressed in human cancers, and many regulatory oncogenic or tumor suppressor small non-coding RNAs (ncRNAs), including microRNAs (miRNAs), have recently been identified to be involved in regulating AKT3 expression. Therefore, a better understanding of regulatory miRNA/AKT3 networks may reveal novel biomarkers for the diagnosis of patients with cancer and may provide invaluable information for developing more effective therapeutic strategies. The aim of this review was to summarize current research progress in the isoform-specific functions of AKT3 in human cancers and the roles of dysregulated miRNA/AKT3 in specific types of human cancers.

## 1. Introduction

AKT, also known as protein kinase (PKB, RAC-PK), is a serine/threonine protein kinase with critical roles in regulating cell signaling in many biological processes. Through phosphorylation of a variety of substrates, AKT regulates the cell cycle and cell proliferation, and inhibits apoptosis via the inactivation of proapoptotic proteins [1]. The aberrant activation or expression of AKT has been reported in many types of human cancers, and has been found to be involved in tumorigenesis, cancer progression, recurrence, and drug resistance [2]. AKT has become an important target for the treatment of human cancer [3]. AKT is activated by a wide range of growth signals, via phosphoinositide 3-kinase (PI3K)-dependent mechanisms. PI3K, a member of the lipid kinase family, is an important regulator of signaling and intracellular vesicular trafficking [4]. Activated PI3K further activates 3-phosphoinositide-dependent protein kinase 1 (PDK1) and AKT by increasing the production of phosphatidylinositol-3,4,5-trisphosphate (PIP3) from phosphatidylinositol 4,5-bisphosphate (PIP2) [5]. Nonetheless, phosphatase and tensin homologue (PTEN), a well-characterized tumor suppressor that inhibits the oncogenic pathway in many cancers, negatively regulates AKT by dephosphorylating PIP3, thereby forming PIP2 [6].

The expression of AKT, in addition to being regulated by proteins, is directly regulated by non-coding RNAs (ncRNAs) [7], which are transcribed from most of the human genome (as much as 80%). Functional ncRNAs, such as microRNAs (miRNAs), circular RNAs (circRNAs), and long ncRNAs (lncRNAs), have garnered research interest because of their versatile roles in fine-tuning signaling pathways involved in diverse biological processes and cancer pathogenesis [8]. MiRNAs are endogenous, highly conserved, and 20–22 nucleotides in length. They block messenger RNA (mRNA) translation or cleavage by binding the 3′-untranslated regions (3′-UTRs) of target mRNAs, including AKT mRNA [7]. The regulatory effects of miRNAs on the PI3K/AKT pathway in human cancers have been reviewed previously [9]. Currently, miRNAs are considered essential players in almost all carcinogenic processes. CircRNAs are a class of single-stranded, covalently closed ncRNAs characterized by their unique structure formed through a 3′–5′ end-joining event [10]. CircRNAs are highly stable, and their expression exhibits spatiotemporal patterns according to cell types, tissues, and developmental stages [11]. They function mainly as competitive endogenous RNAs (ceRNAs) that inhibit miRNA activity, thereby preventing the miRNA-mediated suppression of downstream target gene expression [12]. CircRNAs play essential roles in regulating cellular biological phenotypes and many different pathophysiological processes [13]. Recent studies have revealed that circRNAs are often involved in the PI3K/AKT signaling pathway [14]. Nonetheless, lncRNAs, defined as transcripts longer than 200 nucleotides, regulate gene expression at both the mRNA and protein levels by forming specific RNA:DNA, RNA:RNA, and RNA:protein structures [15]. LncRNA expression is restricted to low copy numbers in different cells at specific developmental stages [16]. LncRNAs either act as ceRNA molecular sponges for miRNAs or serve as functional scaffolds that recruit regulatory proteins to their target chromosomal regions [17,18].

Three AKT isoforms—AKT1 (PKBα), AKT2 (PKBβ), and AKT3 (PKBγ)—are encoded by three different genes that are conserved in mammalian genomes [19]. Despite sharing similar structures (approximately 80% amino acid similarity) [20], AKT isoforms are activated in an isoform-specific manner [21,22], and exhibit different and potentially opposing biological functions, depending on the cellular context. The ablation of individual AKT isoforms exhibits distinct phenotypes in gene knockout mice: Akt1 knockout mice show growth perinatal lethality and elevated apoptosis [23]; Akt2 knockout mice exhibit a diabetes-like phenotype [24]; and Akt3 knockout mice have small brains with mild neurologic defects [25]. All these differences may be caused by the distinct tissue distribution and subcellular localization of AKT isoforms. AKT1 localizes primarily to the cytoplasm, AKT2 localizes to the nucleus, and AKT3 localizes to both the nucleus and the nuclear membrane [26].

The AKT3 encoding gene is located on chromosome 1q44 [27]. Like other AKT members, at the structural level, AKT3 contains an N-terminal pleckstrin homology domain, a central kinase domain (catalytic domain), a C-terminal regulatory domain containing a hydrophobic motif site, and a linker region that tethers the pleckstrin homology domain to the catalytic domain [28] (Figure 1). AKT3 has a more restricted tissue distribution than the other two AKT isoforms, and is primarily expressed in the brain, kidneys, testes, and embryonic heart [29]. The aberrant expression of AKT3 is found in many types of cancer, thus indicating an important role of this isoform in tumorigenesis [30]. In contrast to the abundant evidence supporting the involvement of AKT in cancer induction and progression, relatively limited information is currently available regarding the specific role of AKT3 in oncogenesis. Nonetheless, results have revealed that AKT3 is dysregulated in a variety of cancers, and is associated with abnormal proliferation, apoptosis resistance, and the poor prognosis of tumors [31]. For example, AKT3 promotes the survival of 40–60% non-hereditary melanoma cells and the development of malignant melanoma [32]; in addition, it plays an important oncogenic role in triple negative breast cancer (TNBC), colorectal cancer (CRC), invasion and metastasis of glioma, and lung cancer [2,33,34]. AKT3 may also be involved in tumor angiogenesis by promoting endothelial cell growth, as supported by studies showing that increased endothelial cell proliferation in hemangioma (a benign vascular tumor derived from blood vessel cells) promotes the in vitro angiogenesis of HUVECs as well as in vivo tumor vascularization [35,36]. Herein, we systemically summarize the tumorigenesis-related functions of AKT3 and the regulation of its expression by ncRNAs in specific cancer types.

## 2. AKT3-Derived circRNAs

Remarkably, recent studies have identified several circRNAs derived from the AKT3 gene and their involvement in the regulation of human cancers (Table 1): hsa_circ_0017250, hsa_circ_0112784, hsa_circ_0112781, hsa_circ_0017252, and hsa_circ_0000199 [37]. In particular, hsa_circ_0000199, originating from exons 8–11 of the AKT3 gene, is overexpressed in cisplatin-resistant gastric cancer [38]. Cisplatin treatment is the main chemotherapeutic strategy for hematologic and solid tumor malignancies. Hsa_circ_0000199 enhances cisplatin resistance and gastric cancer (GC) cell survival by promoting the expression of phosphoinositide-3-kinase regulatory subunit 1 (PIK3R1) by sponging miR-198 [38]. Moreover, hsa_circ_0000199 has been found to be highly expressed in lung cancer and to bind miR-516b-5p, thereby regulating glycolysis and decreasing cisplatin sensitivity through the miR-516b-5p/STAT3 axis [37]. In addition, high hsa_circ_0000199 levels are associated with the clinical pathology of TNBC. This circRNA downregulates the expression of the tumor suppressors miR-206 and miR-613, and consequently promotes cell proliferation, migration, and invasion, and facilitates chemo-tolerance in TNBC cells [39]. In contrast, hsa_circ_0017252 is downregulated in renal cell carcinoma (RCC). RCC is one of the most common malignant cancers, and approximately 60–70% of RCC cases are clear-cell RCC (ccRCC). The overexpression of hsa_circ_0017252 suppresses ccRCC metastasis by targeting miR-296-3p, and the miR-296-3p/E-cadherin axis mediates the metastasis inhibitory effect of hsa_circ_0017252 [40]. Another AKT3-derived circRNA, hsa_circ_0017250, arises from the circularization of exons 3–7, whose expression is downregulated in highly malignant glioblastoma multiforme (GBM) and glioma-initiating cells. Hsa_circ_0017250 acts as a tumor suppressor by inhibiting the phosphorylation of AKT, thereby decreasing its activity, and ultimately inhibits glioblastoma cell proliferation and invasiveness [41].

## 3. AKT3 and miRNA/AKT3 Axes in Human Cancers

### 3.1. Breast Cancer

Breast cancer (BC) is among the most common tumors and has the second highest rate of cancer-related death among women. BC tumorigenesis is a multistage process involving several genetic and epigenetic alterations [42]. The distinct functions of AKT isoforms in BC have been reviewed by Basu and Lambring. [22]. Notably, TNBC is an aggressive BC subtype without an available effective targeted therapy. In TNBC, the AKT3 gene is frequently amplified [22,43], and the novel recurrent fusion oncogene MAGI3-AKT3 is enriched [44]. The high level of AKT3 is significantly associated with the duration of patient survival [45]. Moreover, a splice variant of the AKT3 gene whose product lacks a key regulatory Ser472 phosphorylation site induces apoptosis and suppresses TNBC cell growth by upregulating pro-apoptotic Bim and activating Bax and caspase-3 processing [45]. Findings from several functional studies also support the oncogenic roles of AKT3 in BC, as follows. (1) AKT3 has been found to be required for TNBC growth through its downregulation of the cell-cycle inhibitor p27. AKT3 decreases TNBC sensitivity to the pan-Akt inhibitor GSK690693 [46]. (2) AKT3 might stimulate the post-irradiation cell survival of K-RAS-mutated cells after irradiation. Toulany et al. have revealed that AKT3 stimulates the repair of DNA double-strand breaks in oncogenic K-RAS-mutated cells and promotes BC tumor growth in vivo [21]. (3) AKT3 may contribute to endocrine therapy resistance of ErbB2(+) BC cells with aggressive behavior [47], because AKT3 expression and activity are elevated in ErbB2(+) TNBC cells and tamoxifen-resistant BC cells, and activated AKT3 decreases the sensitivity of ErbB2(+) BC cells to tamoxifen (an endocrine therapy used to treat hormone receptor-positive BC).

In contrast to its oncogenic roles, AKT3 has been found to decrease BC cell migration and bone metastasis, but to have no effects on BC tumorigenesis and metastasis [46,48,49,50,51]. Maroulakou et al. have reported that Akt3 gene ablation has no significant inhibitory effects on the development of mammary adenocarcinomas in mouse mammary tumor virus (MMTV)-ErbB2/neu and MMTV-polyoma middle T (PyMT) transgenic mice [48]. Chung et al. have also shown that the knockdown of Akt3 increases cell motility but has no effect on proliferation in mouse BC cells [49]. Beyond these observations in mouse models of BC, Hinz et al. have shown that AKT3 activity is elevated in human bone metastatic MDA-MB-231 cells, and AKT3 may decrease the metastatic potential of these bone-seeking BC cells via the activation of HER2 and discoidin domain receptor (DDR) kinases, and the downregulation of TGFβ [50]. In addition, Lehman et al. revealed that AKT3 promotes the survival of inflammatory BC (IBC, the deadliest form of BC) cells, but has no effect on the invasion of IBC or non-IBC cell lines [51].

In summary, AKT3 does not appear to have pro-oncogenic effects, and it may exert partly contradictory effects, largely depending on the specific BC cell type (Table 2). Nonetheless, emerging studies have indicated that miRNAs have critical roles in BC tumorigenesis by interfering with the PI3K/AKT signaling pathway [52]. The regulation of the AKT3 expression by miRNAs and the functions of the miRNA/AKT3 axes in BC are summarized in Table 3.

Increasing evidence indicates associations between aberrantly expressed miRNAs with BC development and progression, as well as chemoresistance. Several anti-oncogene miRNAs are downregulated in BCs; AKT3 is one of their direct targets, and its expression level is inversely correlated with these miRNAs’ expression levels. These miRNAs include (1) miR-29b, which simultaneously inhibits tumor angiogenesis and tumorigenesis [36]; (2) miR-145, which is downregulated in BC tissues and in docetaxel-resistant BC cells, thus promoting the docetaxel sensitivity of BC cells [53]; (3) miR-29c, which shows the progressive loss of expression during TNBC tumorigenesis and plays a critical role in the early development of TNBC [54]; and (4) miR-181a, which decreases glycolysis in TNBC cells [55]; (5) miR-433, which decreases BC cell proliferation and cell survival [56]; (6) miR-489, which increases BC chemosensitivity, and suppresses BC cell growth and invasion [57]; and (7) miR-3614-3p, which decreases BC tumor cell invasion and migration [58].

In addition to being directly regulated by miRNAs, the AKT3 expression is indirectly regulated by circRNAs and lncRNAs [59,60]. The oncogenic circWHSC1, which is highly expressed in BC tissues and cells, regulates TNBC cell growth, migration, invasion, and survival by sponging the tumor suppressor miR-212-5p, thus promoting AKT3 expression [60]. In BC cell lines and clinical BC tissues, the pseudogene-derived lncRNA RP11-480I12.5 has been found to be overexpressed [59]. Both RP11-480I12.5 and its transcript, RP11-480I12.5-004, exhibit pro-tumorigenic activity by increasing the AKT3 expression by competitively binding miR-29c-3p [59].

**Table 2 cells-12-02594-t002:** Roles of AKT3/Akt3 in breast cancer.

BC Model System	AKT3 or Akt3/Functions	AKT3 Targets	Ref.
K-RAS mutated MDA-MB-231 cells; xenografts	AKT3/cell proliferation↑, tumor growth↑, post-irradiation cell survival↑		[21]
3475 subline of MDA-MB-231 cells (lung metastasis); MDA-MB-231	AKT3/tumor growth↑, metastasis↑, apoptosis↓	ERK, Bim, Bax	[45]
MDA-MB-231; MDA-MB-468 and MCF10DCIS xenografts	AKT3/TNBC growth↑	p27	[46]
ErbB2(+) BC cells, mammary tumor cells	AKT3/cell proliferation↑, tamoxifen sensitivity↓	pErbB2/pErbB3, Foxo3a, ERα	[47]
MMTV-ErbB2, MMTV-PyMT mice (Neu- and PyMT- driven mammary oncogenesis)	Akt3/no effect on tumorigenesis of mouse BC cells		[48]
PyMT mouse BC cells	Akt3/metastasis of mouse BC cells↓		[49]
MDA-MB-231 BO cells; xenografts	AKT3/migration↓, invasion↓ bone metastasis↓	HER2, DDR kinase	[50]
IBC cells: SUM149	AKT3/survival of IBC↑, no effect on invasion		[51]

Note: ↑, increased; ↓, decreased.

**Table 3 cells-12-02594-t003:** The ncRNA and miRNA/AKT3 axes in human cancers.

Tumor Tissue/Cell Lines	LncRNA or circRNA/Functions	MiRNAs in the miRNA/AKT3 Axis/Related Functions	Ref.
Breast Cancer (BC)			
BC tissues; MDA-MB-231, HUVECs		miR-29b/ angiogenesis↓ tumorigenesis↓	[36]
Docetaxel resistance of BC MCF7, MDA-MB-231, docetaxel resistant cell lines: MCF7/DTX, MDA-MB-231/DTX		miR-145/ cell viability↓, colony formation↓, docetaxel sensitivity↑	[53]
MCF10A, MCF10. AT1, MCF10.neoT, CF10. Ca1d, MCF10. Ca1h, MCF10. DCIS		miR-29c/ preneoplastic TNBC cell proliferation↓, colonization ability↓	[54]
MDA-MB-231		miR-181a-5p/ viability↓, migration↓, survival↓, Warburg effect↓	[55]
BC tissues; BT-549, MCF-7, MDA-MB-453, MDA-MB-231		miR-433/ cell proliferation↓, cell viability↓, apoptosis↑	[56]
Drug-resistant and drug-sensitive BC tumor tissues; MCF-7, MDA-MB-231, MDA-MB-468, T47D		miR-489/ chemosensitivity↑, cell proliferation↓, invasion↓	[57]
MCF-7, MDA-MB-231		miR-3614-3p/ invasion↓, migration↓	[58]
BC tissues; MDA-MB-231	circWHSC1/ cell growth↑, proliferation↑, migration↑, invasion↑, glycolysis↑, apoptosis↓	miR-212-5p	[60]
HCC1937, BT549, MDA-MB-231, MCF-7, T47D, BT474	RP11-480I12.5-004/ cell proliferation↑, colony formation↑, apoptosis↓	miR-29c-3p	[59]
Non-small-cell lung cancer (NSCLC)			
NSCLC tissues; cell lines: BEAS-2B, A549, HCC823, NCL-H23, NCL-H358 cells		miR-217/ cell proliferation↓, apoptosis↑	[61]
NSCLC tissues; A549		circulating miR-320a/ metastatic potential↓, apoptosis↑	[62]
NSCLC tissues; NSCLC cell lines: CALU3, CALU6, A549, H1229, H1975	circWHSC1/ colony formation↑, viability↑, metastasis↑, progression↑	miR-296-3p	[63]
NSCLC tissues; NSCLC cell lines: A549 and H460	circ_0016760/proliferation↑, migration↑, apoptosis↓	miR-646	[64]
NSCLC tissues; Cell lines: NCI-H1299, A549, H460, NCI-H2106, H1975	circ_0000520/ cell growth↑, migration↑, invasion↑	miR-1258	[65]
Hepatocellular carcinoma (HCC)			
HCC specimens; HCC cell lines: HepG2		miR-122, miR-124/ function not validated	[66]
HCC tissue samples and cell lines: Huh-7, SNU-182, SNU-475, Hep3B2, HepG2		miR-122/ cell growth↓, migration↓, apoptosis↑,	[67]
HCC-BCLC9 cell		miR-122/ cell proliferation↓, dormancy↑	[68]
HCC specimens; HepG2, HuH7, SMMC-7721		miR-144/ cell proliferation↓, migration↓, invasion↓	[69]
HCC tissues of solitary large, nodular, and small HCC; HCC cell lines: SMMC7721, HepG2, HUH7, MHCC97-L, MHCC97-H, HCCLM3		miR-424/ cell proliferation↓	[70]
HCC specimens; HCC cell lines: QGY-7703, Huh7, BEL-7402, HepG2, Hep3B		miR-582-5p/ colony formation↓, cell proliferation↓	[71]
HCC tissues; HCC cell lines: SNU-449, SNU-182, Huh7, LM3, Bel-7405, SK-hep1, Hep3B	LINC00680/stemness behavior↑, chemosensitivity↓	miR-568	[72]
HCC specimens; HCC cell lines: HepG2, Hep3B, Huh-7, SNU398, NU449, SNU182, SNU475		miR-519d/AKT3? miR-519d/ cell proliferation↑, migration↑, apoptosis↓	[73]
Colorectal cancer (CRC)			
CRC cell lines: RKO, HCT116		miR-124-3p.1/ proliferation↓, metastasis↓	[74]
CRC tissues; CRC cell lines: SW480, HCT116, LOVO, SW620		miR-384/ proliferation↓	[75]
CRC tissues; CRC cell lines: Colo205, SW620, HCT116, HT29, LOVO, SW480	LINC02163/proliferation↑, metastasis↑	miR-511-3p	[76]
CRC tissues; CRC cell lines: LOVO, PKO, SW480, HT29	lncRNA DSCAM-AS1/proliferation↑, invasion↑, migration↑	miR-384	[77]
Gastric carcinoma (GC)			
GC tissues; cell lines: SGC-7901, MKN45, BGC823		miR-195/ apoptosis↑	[78]
Gastric adenocarcinoma serum; cell line: MGC-803	MALAT1/ apoptosis↓	miR-181a-5p	[79]
GC tissues; GC cell lines: MKN28, NCI-N87, AGS, KATOIII, RF1, RF48	circNF1/cell proliferation↑	miR-16	[80]
Cholangiocarcinoma (CCA) cell lines: HCCC-9810, RBE	circRNA CDR1a/proliferation↑, invasion↑	miR-641	[81]
Pancreatic cancer (PC) tissues; cell line: PANC-1		miR-489/proliferation↓, apoptosis↑	[82]
Ovarian cancer (OC)/epithelial ovarian cancer (EOC)			
EOC tissues; cell lines: SKOV3, A2780, HO8910, 3AO		miR-29b/ Warburg effect↓, tumor growth↓	[83]
OC cell lines: SKOV3, OVCAR3, cisplatin-resistant SKOV and OVCAR3 cells		miR-489/ survival↓, growth↓, apoptosis↑, sensitivity of cisplatin-resistant OC to cisplatin↑	[84]
OC tissues; Cell lines: CaOV3, OVCAR3, SKOV3	RHPN1-AS1/proliferation↑, migration↑, invasion↑	miR-665	[85]
OC tissues; OC cell lines: SKOV-3, ES-2, OVCAR3, A2780, CAOV3	lncRNA EMX2OS/proliferation↑, invasion↑, tumor growth ↑	miR-654	[86]
Endometrial carcinoma (EC)			
EC tissues; EC cell line ECC1		miR-582-5p/proliferation↓, apoptosis↑	[87]
Endometrial adenocarcinoma cell line: Ishikawa (ISK) cells	lncCDKN2B-AS1/proliferation↑, invasion↑	miR-424-5p	[88]
EC tissues; EC cell lines: HEC1A, HEC1B, Ishikawa	LINC01224/ proliferation↑, apoptosis↓	miR-485-5p	[89]
Thyroid carcinoma (TC)/papillary thyroid carcinoma (PTC)			
TC tissues; cell lines: TPC-1, FTC-133, 8505C; primary PTC cells;		miR-145/ growth↓, metastasis↓	[90]
PTC tissues; PTC cell line: K1		miR-29a/ growth↓, apoptosis↑, metastasis↓	[91]
TC tissues; PTC cell lines: 8505C, TPC-1, SW1736		miR-217/ proliferation↓, migration↓, invasion↓	[92]
PTC tissues; PTC cell lines: B-CPAP, KTC-1	lncRNA n384546/progression↑, metastasis↑	miR-145-5p	[93]
TC tissues; TC cell lines: BCPAP, K1, H7H83, TPC-1	circ_0000144/proliferation↑, migration↑, invasion↑	miR-217	[94]
Nasopharyngeal carcinoma (NPC)			
NPC tissues; Human primary NPC cell		miR-424-5p/ proliferation↓, migration↓, apoptosis↑	[95]
NPC tissues; NPC cell lines: C666-1, SUNE1, 5-8 F, HNE1, HNE2	circTRAF3/proliferation↑, invasion↑, apoptosis↓	miR-203a-3p	[96]
Oral squamous cell carcinoma (OSCC) tissues; cell line: SCC-4, SCC-25, HN-6, CAL-27, TCA-83		miR-16/ proliferation↓, apoptosis↑	[97]
Glioblastoma multiforme (GBM)			
GBM cell lines: T98G, U87, A172, LN229, LN18		miR-610/ proliferation↓, anchorage independent growth↓	[98]
GBM cell lines: LN229, A172, U373, SHG44	lncRNA, GAS5/proliferation↓, migration↓, invasion↓	miR-424	[99]
Multiple myeloma (MM)			
Primary MM cells, MM cell lines: MM.1S, RPMI8226		miR-15a, miR-16-1/ cell proliferation↓	[100]
MM cell lines: OPM2, RPMI-8226; Endothelial cell: HUVECs		miR-29b/ endothelial cell proliferation↓, migration↓, tube formation↓	[101]
MM tissues; cell lines: KM3, H929, MM1S, U266 cells	circ_0000142/proliferation↑, metastasis↑	miR-610	[102]
MM tissues; cell lines: OPM-2, U266, KM3, U1996, H929	lncRNA FEZF1-AS1/ proliferation↑, apoptosis↓	miR-610	[103]
Osteosarcoma (OS)			
OS tissues; cell lines: HOS, MG-63, Saos-2, SW1353, U2OS		miR-1258/ proliferation↓	[104]
OS tissues; OS cell lines: HOS, MG-63, SaOS-2, U2OS,	MALAT1/ glycolysis↑, proliferation↑, metastasis↑	miR-485-3p	[105]
Uveal melanoma (UM) tissues; UM cell line: OCM-1A		miR-224-5p/ proliferation↓, migration↓, invasion↓	[106]
UM cell lines: OMM2.5, UM001, Mel285, Mel290; UM xenografts		miR-181a-5p/ proliferation↓, colony formation↓, apoptosis↑, tumor growth ↓	[107]
NK/T cell lymphoma (NKTL) tissues; cell lines: KHYG-1, NK-92, HANK-1, SNK-1, SNK-6		miR-150/ sensitivity of NKTL to radiation treatment↑	[108]
Bladder cancer TCGA database		miR-195/ cell proliferation↓	[109]
Wilms’ tumor tissues; cell lines: 17–94, WIT49		miR-22-3p/ proliferation↓, invasion↓	[110]

Note: ↑, increased; ↓, decreased.

### 3.2. Lung Cancer

Lung cancer is the second most common cancer worldwide and the leading cause of cancer death. Approximately 80–85% of lung cancer is non-small-cell lung cancer (NSCLC). AKT3 has been found to have high activity in NSCLC cells, thereby promoting proliferation, survival, and migration [111]. Emerging studies indicate that AKT3 expression is regulated by several miRNAs that are downregulated in NSCLC (Table 3). People with NSCLC with lower levels of miR-217 have a shorter overall survival. The overexpression of miR-217 inhibits NSCLC cell proliferation and induces apoptosis [61]. Circulating miR-320a levels are relatively low in the plasma of people with NSCLC; these low levels are correlated with clinicopathological characteristics, such as tumor size, tumor stage, and lymph node metastasis. Circulating miR-320a functions as a tumor-suppressor miRNA that decreases metastatic potential and increases the apoptosis of NSCLC cells [62]. Notably, three circRNAs are upregulated in NSCLC and consequently promote NSCLC cell growth and metastasis: namely circWHSC1, circ_0016760, and circ_0000520. These circRNAs act as ceRNAs, thereby increasing the AKT3 expression by directly targeting miR-296-3p, miR-646, and miR-1258, respectively, in NSCLC cells [63,64,65].

### 3.3. Digestive/Gastrointestinal Cancers

Hepatocellular carcinoma (HCC), the most frequently occurring liver malignancy, has high rates of fatality, recurrence, and chemotherapeutic resistance. The deregulation and activation of the AKT signaling pathway is common in HCC and is associated with poor patient prognosis. AKT3 expression is upregulated in the HCC cell lines SNU-182, Hep3B2, and SNU-475 [67]. Furthermore, a growing number of HCC-associated genes are being found to be regulated by ncRNAs. For example, a panel of serum miRNAs comprising miR-26a-5p, miR-122-5p, miR-141-3p, miR-192-5p, miR-199a-5p, miR-206, miR-433-3p, and miR-1228-5p has shown clinical value in HCC diagnosis [112]. The oncogenic lncRNA LINC00680 is markedly upregulated in HCC tissues. LINC00680 acts as a ceRNA sponging miR-568, thus activating AKT3, enhancing HCC stemness behavior, and decreasing the chemosensitivity to 5-fluorouracil (a highly effective classical chemotherapeutic agent in the treatment of HCC) [72]. Moreover, Zhang et al., in a comprehensive analysis of the miRNA-regulated protein interaction network, identified a list of miRNAs targeting AKTs, which includes miR-122/AKT3 and miR-124/AKT3 interactions, thus suggesting the critical roles that these miRNAs in HCC malignant progression [66]. Further studies have validated that several tumor suppressor miRNAs that directly suppress AKT3 expression are lost or downregulated in HCC cells (Table 3). MiR-122, the most highly expressed miRNA in the healthy adult liver, is responsible for liver stem cell differentiation towards the hepatocyte lineage. Its expression is frequently lost in HCC tissues and HCC cell lines [67,68]. By binding the 3′-UTR of AKT3 and controlling AKT3 gene expression, miR-122 inhibits HCC cell proliferation, increases the chemosensitivity of HCC cells, and attenuates HCC tumor growth in vivo [67,68]. The forced overexpression of miR-144, miR-424, or miR-582-5p in HCC cells results in anti-HCC effects, by suppressing cell proliferation, migration, invasion, or survival [69,70,71]. Notably, miR-519d is upregulated in HCC and contributes to hepatocarcinogenesis after anticancer treatments [73]. Although AKT3 is known to be directly targeted by oncogenic miR-519d, the pathophysiological role of AKT3 in HCC remains unclear.

CRC is a common digestive malignancy in females and males worldwide. The initiation and malignant progression of CRC is a long-term multi-stage process involving genetic and epigenetic alterations. The oncogenic GLUT5 expression in tumor tissue is associated with aggressive behavior and invasiveness in cancer cells, and also regulates the migratory activity in drug-resistant CRC cells [113]. In CRC, AKT3 contributes to the drug resistance by aberrantly downregulating miR-125b-5p expression, thus leading to the expression of the glucose transporter GLUT5. In addition, the AKT3 expression in CRC cells is regulated by miR-124-3p.1 [74] and miR-384 [75]. Both miR-124-3p.1 and miR-384 exhibit anti-CRC effects by inhibiting the cell proliferation [74,75]. Moreover, two lncRNAs, namely LINC02163 and Down syndrome cell adhesion molecule antisense1 (DSCAM-AS1), have shown oncogenic effects in CRC development and progression [76,77]. Their expression is upregulated in CRC and is associated with tumor metastasis and poor prognosis in people with CRC. LINC02163 and DSCAM-AS1 function as ceRNAs that target miR-511-3p and miR-384, respectively, and subsequently induce the AKT3 expression [76,77] (Table 3).

GC, the third leading cause of cancer deaths worldwide, is usually diagnosed in advanced stages. The miR-195/AKT3 axis has a critical role in GC development. The transcription factor early growth response 1 (EGR1) functions as an oncogene in GC by suppressing the apoptosis of GC cells by directly inhibiting the expression of miR-195 and activating AKT3 [78]. Metastasis-associated lung adenocarcinoma transcript 1 (MALAT1), also known as non-coding nuclear-enriched abundant transcript 2 (NEAT2), promotes gastric adenocarcinoma through the MALAT1/miR-181a-5p/AKT3 axis [79]. High levels of MALAT1 are detected in the serum in people with gastric adenocarcinoma. MALAT1 regulates GC cell proliferation and apoptosis by decreasing the expression of miR-181a-5p, which in turn upregulates AKT3 protein levels [79]. On the basis of its high expression in GC tissues and cell lines, circNF1 has been identified as an oncogenic circRNA in GC, which promotes GC cell proliferation, binds miR-16, and consequently derepresses its downstream target AKT3 [80].

Cholangiocarcinoma (CCA, also known as bile duct cancer), is one of the most common hepatic malignancies and accounts for 3% of all gastrointestinal cancers. CCA is difficult to diagnose in early stages, and, because of early metastasis, its 5-year survival rates are only 20–40%. The upregulated circRNA cerebellar degeneration-associated protein 1 antisense (CDR1as, also known as cirs-7) has oncogenic roles in CCA. CDR1as binds miR-641, and subsequently accelerates miR-641 degradation and may possibly lead to the upregulation of AKT3 expression [81].

Pancreatic cancer (PC) is a high-grade malignancy of the digestive system with vague clinical features in early stages. The 5-year overall survival rate of PC is extremely low, at less than 5%. The level of miR-489 is markedly low in PC. The upregulation of miR-489 significantly inhibits cell proliferation and induces the cell apoptosis of PC PANC-1 cells by targeting AKT3 [82].

### 3.4. Gynecologic Cancers

Ovarian cancer (OC) is the most lethal gynecological cancer, accounting for 5% of cancer cases and deaths among females worldwide. Epithelial ovarian cancer (EOC) accounts for ~90% of all ovarian malignancies. In EOC cells, the AKT3 expression is negatively regulated by downregulated miR-29b, thus inducing the rate-limiting glycolytic genes hexokinase 2 and pyruvate kinase M2 and increasing the Warburg effect and ovarian cancer progression [83]. A role of AKT3 in the cisplatin resistance of OC cells has been suggested, because MK-2206 2HCl, an AKT3 inhibitor, increases the sensitivity of cisplatin-resistant OC cells to cisplatin [84]. However, miR-489, which regulates AKT3 expression, is downregulated in OC cells. The miR-489 overexpression increases the sensitivity of cisplatin-resistant OC cells to cisplatin by inhibiting cell growth and promoting apoptosis [84]. The expression of the oncogenic lncRNA RHPN1-AS1, a 2030 bp transcript from human chromosome 8q24, is upregulated in OC tissues and cell lines. RHPN1-AS1 promotes OC cell proliferation and migration via the miR-665/AKT3 axis [85]. Another oncogenic lncRNA, EMX2OS, is overexpressed in OC tissues, and directly suppresses miR-654 expression, thus leading to the upregulation of AKT3; therefore, the EMX2OS/miR-654/AKT3 axis may confer aggressiveness in OC [86].

Endometrial carcinoma (EC) is a common cancer of the female genital tract. Invasion and recurrence contribute to the prognosis and survival rate of patients with EC. AKT3 is involved in the regulation of cell proliferation and the invasion of endometrial stromal cells [88]. AKT3 expression is regulated by the tumor inhibitor miR-582-5p, whose expression is significantly diminished in human EC tissues. MiR-582-5p strongly inhibits cell proliferation and promotes the apoptosis of EC cells [87]. The oncogenic lncRNA CDKN2B-AS1 has been observed in Ishikawa endometrial adenocarcinoma cells and found to promote cellular proliferation and invasion by sponging miR-424-5p, thus upregulating the expression of AKT3 [88]. High levels of the lncRNA LINC01224 have been found in both EC tumor tissue and cell lines. LINC01224 promotes EC cell proliferation and inhibits apoptosis by elevating the expression of AKT3 by targeting miR-485-5p [89].

### 3.5. Thyroid Carcinoma and Head and Neck Cancers

Papillary thyroid cancer (PTC), the most common histological type of thyroid carcinoma (TC), accounts for more than 85% of TC cases. Environmental exposure and genetic mutation are the risk factors for PTC. MiR-145, miR-29a, and miR-217 are downregulated in TC/TPC. Their tumor suppressor roles have been demonstrated to involve the direct inhibition of AKT3 in PTC cells [90,91,92]. In agreement with these findings, miR-145-5p is a key miRNA target of the oncogenic lncRNA n384546, whose expression is elevated and associated with clinicopathological features in patients with PTC. Moreover, n384546 promotes TPC progression and metastasis by acting as a ceRNA of miR-145-5p, thereby regulating AKT3 [93]. In addition, circ_0000144, an oncogenic circRNA, is markedly elevated in TC tissues and is associated with tumor size, TNM stage, and lymph node metastasis in people with TC. Circ_0000144 shows cancer-promoting effects on TC cells by regulating the miR-217/AKT3 pathway [94].

Nasopharyngeal carcinoma (NPC) is a common head and neck cancer with a high incidence in southern China, North Africa, and Southeast Asia. As with most tumors, genetic abnormalities are closely associated with the occurrence of NPC. MiR-424-5p is a tumor-associated miRNA encoded at human Xq26.3. In NPC, miR-424-5p expression is downregulated, and is associated with lymph node metastasis and clinical staging. MiR-424-5p exhibits anti-oncogenic activities by inhibiting the proliferation, migration, and invasion of NPC cells by decreasing the AKT3 expression [95]. The circRNA circTRAF3 is highly expressed in NPC and is associated with metastasis and survival in people with NPC. CircTRAF3 promotes NPC cell proliferation and metastasis by eliminating the inhibitory effect of miR-203a-3p on AKT3 expression [96].

Oral squamous cell carcinoma (OSCC) is the most prevalent subgroup of head and neck cancer and the most common type of oral cancer. AKT3 is directly targeted by miR-16, and its expression is negatively associated with that of miR-16 in OSCC. MiR-16 is dysregulated in OSCC [97], and it functions as a tumor suppressor miRNA, thus inhibiting cell proliferation and inducing apoptosis in OSCC by decreasing AKT3 [97].

### 3.6. Other Types of Human Cancer

GBM is the most common and lethal primary brain malignancy. People with GBM have poor prognosis and survival. AKT activation is found in approximately 80% of human GBMs. AKT3 is overexpressed in glioma cells and has been found to play a critical role in GBM [114]. Studies have shown that the siRNA knockdown of AKT3 in GBM T98G cells significantly decreases cell viability, proliferation, invasion, and metastasis [34]. MiR-610 is downregulated in GBM, and it directly suppresses AKT3 expression, thus decreasing the proliferation and anchorage-independent growth of GBM cells [98]. Interestingly, lncRNA growth arrest-specific 5 (GAS5) functions as a tumor suppressor in glioma cells by alleviating the promoter methylation of miR-424, and consequently increasing the expression of miR-424, and suppressing AKT3 and its targets, cyclinD1, c-Myc, Bax, and Bcl-2 [99].

Multiple myeloma (MM), the second most common hematological malignancy, is characterized by high infiltration and the multifocal proliferation of malignant plasma cells in the bone marrow. The miR-15a/miR-16-1 cluster resides at chromosome 13q14, an area frequently deleted in MM [115]. The decreased expression of miR-15a and miR-16-1 may be involved in the pathogenesis and progression of MM, because both these miRNAs inhibit cell proliferation and suppress AKT3 expression in MM cells [100]. The miR-29b/AKT3 axis was found to be involved in the progression of MM by regulating the angiogenic activity of MM-derived exosomes [101]. In endothelial cells, after treatment with exosomes released from MM cells treated with C6-ceramide, the expression of the tumor suppressor miR-29b is induced, but AKT3 is decreased; consequently, endothelial cell proliferation, migration, and angiogenesis are suppressed [101]. MiR-610 functions as a tumor suppressor and suppresses AKT3 in several human cancers. Both circ_0000142 and lncRNA FEZF1-AS1 are upregulated and associated with poor prognosis in people with MM and promote MM cell growth by modulating the miR-610/AKT3 axis [102,103].

OS, an extremely malignant primary bone cancer with rapid progression, affects both children and adolescents. The prognosis of OS is poor, due to its strong tendency towards lung metastasis. AKT3 is upregulated in OS and is associated with OS progression; its expression is negatively associated with the expression of miR-1258 and miR-485-3p in OS tissues [104,105]. MiR-1258 has been widely studied in various cancers including OS. The upregulation of miR-1258 significantly inhibits OS cell growth by binding the 3′-UTR of AKT3 [104]. Nonetheless, the anti-oncogene miR-485-3p, with inhibitory effects on glycolysis and metastasis of OS cells, is downregulated in OS by the lncRNA MALAT1 [105].

Uveal melanoma (UM) is a rare intraocular malignancy in adults, which arises from melanocytes in the iris, ciliary body, or choroid. In UM tissues, the miR-224-5p expression is low, whereas the AKT3 expression is high. MiR-224-5p is involved in the proliferation, invasion, and migration of UM cells, partially through the regulation of the expression of AKT3 [106]. Of note, our research team has recently discovered that targeting AKT3 in UM cells might be a mechanism underlying the inhibitory effect of miR-181a-5p on UM development [107].

NK/T cell lymphoma is a rare non-Hodgkin lymphoma with high invasive malignancy, putative NK-cell origin, and poor prognosis. MiR-150 increases the sensitivity of NK/T cell lymphoma to ionizing radiation through direct targeting AKT3, but it is significantly diminished in NK/T cell lymphoma tissues and cell lines [108].

Bladder cancer is a relatively rare, highly malignant tumor arising from urinary bladder tissues. AKT3 may participate in the proliferation and apoptosis of bladder cancer cells, and it is an important target of miR-195. MiR-195 has anti-cancer roles by suppressing glucose uptake and the proliferation of bladder cancer cells [109].

Wilms’ tumor, or nephroblastoma, the most frequent renal cancer in children, occurs primarily in the first 5 years after birth. AKT3 is upregulated in Wilms’ tumors, whereas miR-22-3p is downregulated. MiR-22-3p regulates the proliferation and invasion of Wilms’ tumor cells through the inhibition of AKT3 [110].

## 4. Dysregulation of the Same miRNA/AKT3 Axis in Different Cancers

As discussed above, in a specific type of human cancer, different dysregulated ncRNA/AKT3 axes are involved in tumorigenesis, progression, and drug resistance (Table 3). However, studies have also identified the same disrupted miRNA/AKT3 axis in different cancer types (Figure 2). For instance, the miR-145/AKT3 axis is found in several human cancers including BC, chemo-resistant BC, and PTC [53,90,93]. Furthermore, the miR-181a-5p/AKT3 axis plays important roles in gastric adenocarcinoma, TNBC, and UM [55,79,107]. In addition, the miR-217/AKT3 axis is dysregulated in TC and NSCLC [61,92,94]; the miR-424-5p/AKT3 axis is dysregulated in HCC, NPC, endometrial adenocarcinoma, and glioma [70,88,95,99]; the miR-489/AKT3 axis is dysregulated in drug-resistant BC, cisplatin-resistant OC cells, and PC [57,82,84]; and the miR-610/AKT3 axis is dysregulated in GBM and MM [98,102,103]. Notably, the miR-29 family consists of miR-29a, miR-29b-1, miR-29b-2, and miR-29c, which are involved in many pathophysiological processes, and are associated with cancer development and metastasis [116]. The tumor suppressor role of miR-29b has also been reviewed [117]. Abnormal miR-29/AKT3 axes are found in certain types of human cancer, e.g., miR-29a/AKT3 in thyroid cancer [91]; miR-29b/AKT3 in BC, EOC, and MM [36,83,101]; miR-29c/AKT3 in TNBC [54]; and miR-29c-3p/AKT3 in BC [59]. Together, these findings suggest that a single miRNA may be used as an efficient anti-cancer therapeutic agent for different types of human cancer [36].

Although many miRNAs have been found to regulate the expression of AKT3 in cancer cells, much less is known regarding the underlying mechanisms of the miRNA/AKT3 axis in cancer progression, metastasis, and drug resistance (Figure 3). Recently, programmed death-ligand 1 (PD-L1) has been identified as the downstream mediator of the miR-654-3p/AKT3 axis in regulating the proliferation and invasion of ovarian cancer cells [86]. PD-L1 is a key immune checkpoint molecule with critical roles in evading anti-tumor immunity [118,119]. In addition, the miR-568/AKT3 axis modulates the phosphorylation of its downstream signaling molecules, including mTOR, elF4EBP1, and p70S6K, thereby regulating HCC stemness and chemosensitivity [72]. Moreover, the miR-424/AKT3 axis plays roles in glioma progression and invasion by altering the expression of cyclinD1, c-Myc, Bax, and Bcl-2 [99]. Given the interesting emerging roles of AKT3 in human cancers, important mechanistic insights regarding the AKT3 downstream effectors are expected to be discovered in the future.

## 5. Perspectives and Therapeutic Applications

Despite substantial efforts to develop effective strategies, the outcomes for patients with highly malignant cancer remain unsatisfactory. Studying the potential mechanisms involved in the occurrence and progression of aggressive cancers is crucial to explore novel targets for disease diagnosis and treatment. Given the central role of AKT signaling in the pathogenesis and development of cancer, discovering the exact mechanisms controlling AKT signaling activity, and understanding how the inhibition of this pathway influences major cellular processes, is critical. Exciting findings have recently indicated that ncRNAs function through molecular associations with the components of classical signaling pathways including the AKT signaling pathway. Many ncRNAs involved in this pathway have been identified and extensively studied. Dysregulated miRNAs and other ncRNAs are gradually becoming accepted as biomarkers in cancer diagnosis and as potential targets in cancer therapy. In particular, the miRNA/AKT3 axis is attracting increasing attention.

Currently available prediction tools and databases, together with microarray techniques and second-generation sequencing, readily enable the examination of groups of ncRNAs with sequence complementarity to the 3′-UTR of AKT3. By integrating the ncRNA profile and miRNA target prediction tools, links between ncRNAs and AKT3 can be established and validated in cancer cells. Hence, many miRNAs that directly bind the 3′-UTR of AKT3 and participate in the regulation of AKT signaling in human cancers have been revealed, and additional AKT3-associated ncRNAs are expected to be identified in the near future. Moreover, AKT3-derived circRNAs target some miRNAs in certain types of cancers, thus suggesting that AKT3 might regulate epigenetic modifiers in a post-transcriptional manner. However, limited information is available regarding AKT3-driven miRNA alterations. The findings regarding miRNA/AKT3 regulatory networks may revolutionize views regarding the genesis, progression, and treatment of cancer. The use of ncRNAs in cancer diagnosis has rapidly increased, particularly that of miRNAs [120], because of their tissue specificity and stability in the circulating blood [121,122]. The identification of important miRNAs is expected to aid in the discovery of novel cancer specific biomarkers and therapeutic targets.

Because the upregulated expression or activation of AKT3 is frequently observed in a wide variety of human cancers, and is associated with cancer progression and chemotherapy resistance, the specific targeting of AKT3 may be attractive in new drug discovery. Unfortunately, to date, no AKT3-specific inhibitor is available. Although the generation of AKT3-specific nanobodies has been described and characterized in vitro assays, in vivo functional validation remains lacking [123]. Alternatively, AKT3 can be regulated by modifying the ncRNA expression for cancer therapy. The downregulation of AKT3 can be achieved by increasing tumor suppressor miRNAs or by suppressing oncogenic miRNAs. Numerous miRNAs directly target AKT3 (Table 2), whereas restoring their expression decreases AKT3 expression and exhibits therapeutic effects against a variety of cancers. Thus, regulating ncRNAs in the miRNA/AKT3 axis to control cancer may be a therapeutic option. For developing drugs that stably regulate ncRNA activity and efficiently elicit effects, in-depth studies of the structures and functions of miRNAs associated with the AKT3 pathway will be essential. Notably, the development of miRNA-based therapeutics with nanotechnology-based delivery systems may enable the more effective targeting of cancer cells, and may aid in translating miRNA-based cancer therapeutics into clinical approaches. For instance, a miR-181a-5p mimic encapsulated in lipid nanoparticles and in hyaluronan decorated lipid nanoparticles has displayed strong anti-neoplastic effects in retinoblastoma and glioblastoma, respectively [124,125]. The co-delivery of both miRNAs and small molecule drugs by nanoparticles can achieve additive anticancer therapeutic effects [125].

## 6. Conclusions

Given the growing evidence indicating the oncogenic roles of AKT3, further studies should investigate how AKT3 exerts its effects at the molecular level, and AKT3 should be subjected to drug discovery for specific molecular inhibitors. Many ncRNAs regulate the expression of AKT3, and consequently cancer development, progression, and chemosensitivity. Knowledge regarding the roles of the dysregulated miRNA/AKT3 axis in oncogenic signaling pathways could be advanced to enable the diagnosis of specific cancer types. Analyzing miRNA reprogrammed AKT3 profiles and AKT3 regulated miRNA profiles to explore cancer pathogenesis in a personalized or cancer-specific manner may advance the development of effective new therapeutic strategies in the future. A comprehensive investigation of the roles of the miRNA/AKT3 axis in cancer-associated pathways and their clinical relevance in different cancers will ideally enable combinatorial therapeutic strategies comprising miRNA overexpression and the administration of specific inhibitors or AKT3 specific inhibitors.

## Figures and Tables

**Figure 1 cells-12-02594-f001:**
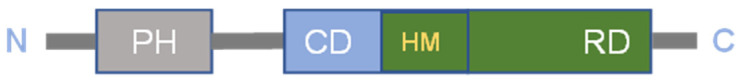
Structure of human AKT3. PH, pleckstrin homology-domain; CD, catalytic domain; RD, regulatory domain; HM, hydrophobic motif.

**Figure 2 cells-12-02594-f002:**
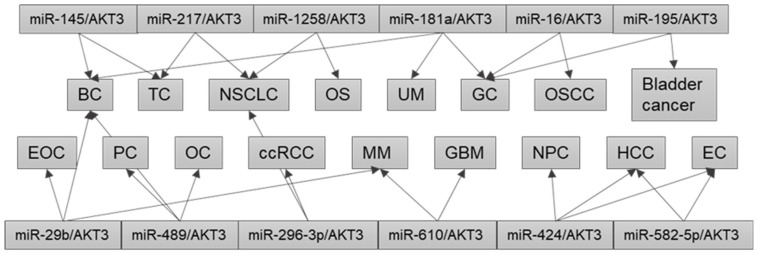
Dysregulation of the same miRNA/AKT3 axis in different cancers. BC, breast cancer; ccRCC, clear cell renal cell carcinoma; EC, endometrial carcinoma; EOC, epithelial ovarian cancer; GBM, glioblastoma multiforme; GC, gastric cancer; HCC, hepatocellular carcinoma; MM, multiple myeloma; NPC, nasopharyngeal carcinoma; NSCLC, non-small-cell lung cancer; OC, ovarian cancer; OS, osteosarcoma; OSCC, oral squamous cell carcinoma; PC, pancreatic cancer; TC, thyroid carcinoma; UM, uveal melanoma.

**Figure 3 cells-12-02594-f003:**
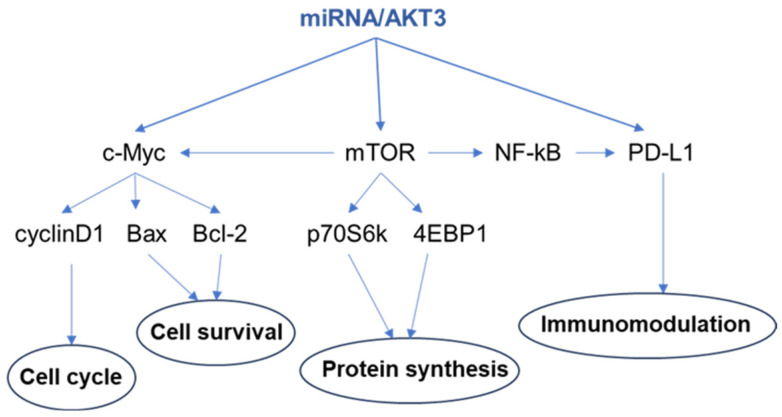
Identified signaling molecules involved in the miRNA/AKT3 axis in human cancers.

**Table 1 cells-12-02594-t001:** AKT3-derived circRNAs and their effects in human cancers.

Tumor Tissue/Cell Lines	CircAKT3/Functions	Targets	Ref.
Lung cancer tissues; cell lines: A549 and H1299	hsa_circ_0000199/ glycolysis↑, cell growth↑, drug sensitivity↓	miR-516b-5p, STAT3	[37]
GC tissues; cell lines: SGC7901, BGC823, CDDP-resistant SGC7901, CDDP-resistant BGC823	hsa_circ_0000199/ DNA damage repair↑, cell survival↑	miR-198	[38]
TNBC tissues; cell lines: MCF-10A, MDA-MB-231, MDA-MB-468, SK-BR-3	hsa_circ_0000199/ chemo-tolerance↑, proliferation↑, migration↑, invasion↑	miR-206, miR-613	[39]
ccRCC tissues; cell lines: OSRC-2, Caki-1, SN12-PM6, A498, SW839	hsa_circ_0017252/ metastasis↓	miR-296-3p	[40]
GBM tissues; cell lines: U251, HS683, SW1783, U373, glioma-initiating cells	hsa_circ_0017250/ proliferation↓, invasiveness↓	AKT	[41]

Note: ↑, upregulated; ↓, downregulated.

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
