# Peer review of "The Multifunctional Nature of the MicroRNA/AKT3 Regulatory Axis in Human Cancers"

_cells, 2023, doi:10.3390/cells12222594_

Round 1

Reviewer 1 Report

Comments and Suggestions for Authors

The review article from Yang and Hardy "The multifunctional nature of the microRNA/AKT3 regulatory axis in human cancers" discusses the role of regulatory RNAs like lncRNAs or mirRNAs within the AKT network. The authors more or less describe different cancer types in the light of this molecular pathway.

One the one hand this is quiet informative, however the article lacks critical discussion with the potential role of regulatory RNAs in cancer therapy. The authors should also include more background information on the underlying molecular mechanisms.  

In fact table 3 is very interesting, however the text lacks critical discussion and just summarises the information of the table.

The manuscript also lacks a final conclusion. In line with this figure 2 is very confusing and does not really provide a summary of the finding, as I would expect from a critical review article.

Author Response

Responses:

We sincerely appreciate the valuable comments and suggestions from the reviewer. We have closely followed the suggestions and comments and made revisions accordingly.

As suggested by the reviewer, the discussion with the potential role of regulatory RNAs in cancer therapy is added in this revised manuscript on page 14 and page 15. We also added a figure (Figure 3) that generally depicting the underlying molecular mechanisms and more discussion on page 14.  A final conclusion is added on page 15.

Reviewer 2 Report

Comments and Suggestions for Authors

The author need to add at least 3 molecular mechanism figures in microRNA/AKT3 regulatory pathway in human cancers.

The author need to write the following points to write before conclusion:

5. Perspectives and therapeutic application of microRNA/AKT3 regulatory axis in human cancers

6. Current drug target and treatment of microRNA/AKT3  human cancers

7. Limitations and overcome of microRNA/AKT3 regulatory axis in human cancers

Added these paragraph with recent references.

Comments on the Quality of English Language

Need to revise English ones more.

Author Response

Responses:

We highly appreciate the reviewer’s insightful comments and constructive suggestions on our manuscript.

As reviewer suggested, we have added an illustrative figure of molecular mechanism (Figure 3) and related information on page 14.

The perspectives and therapeutic application, current drug target and treatment, and limitations and overcome of miRNA/AKT3 regulatory axis in human cancers, have been discussed and added with recent references in Section 5 (Perspectives and therapeutic application) on page 14 and page15.

English Language is revised by the editors of Science Journal (see attached certificate).

Reviewer 3 Report

Comments and Suggestions for Authors

Article n° CELLS - 2693205

Title: The multifunctional nature of the microRNA/AKT3 regulatory 2 axis in human cancers

Authors:  Yang C , Hardy P.

General comment

Yang and Hardy a review focused on the specific modulatory action of mRNAs on the AKT3 networks involved in aberrant cell survival mechanisms associated with tumorigenesis, metastasis, and chemoresistance. As stated by authors, this collection of studies may reveal novel biomarkers for diagnosis of patients with cancer and provide important information for developing more effective therapeutic strategies.

The manuscript is almost interesting and well-organised and it should be of interest to scientist working in the field of the biochemistry of cancer as well as other s with closely related research interest. Therefore, this article collects the novelty of interest for the research community and  in conclusion, considering the above matters, I retain that this manuscript could merit to be accepted for publication in Cells. However I have some suggestions to give more value to the manuscript and to attract further interest of the reader.

1)      The following mRNAs: mir-150, mir-15a, mir-16-5p and mir-16 are involved directly or indirectly with AKT3. Why didn't the authors report anything about it? I believe some of them should be mentioned for their effects on cancer cells.

2)      What role does the NF-kB factor play in the control by mRNAS and AKT3? This evidence also appears to be important in the regulation of cancerous development.

3)      It would also be appropriate to add a paragraph relating to innovative and future pharmacological perspectives on the use of these mRNAS.

     4)      It would be appropriate to include figures that generally show the  biochemical pathways involved in the development of cancer and the modulating action of mRNAs.

Comments on the Quality of English Language

There are some typographical errors

Author Response

Responses:

We appreciate the positive feedback from the reviewer. We thank the reviewer helpful suggestions and made corrections in the revised manuscript. 

  • MiR-150 increases the sensitivity of NK/T cell lymphoma to ionizing radiation through direct targeting AKT3[1], which is mentioned on page 12. The involvement of miR-16/AKT3 in gastric cancer (GC) [2] and Oral squamous cell carcinoma (OSCC)[3] is mentioned on page 10 and page 12 respectively. The information of miR-15a/miR-16-1 involvement in multiple myeloma (MM) and targeting AKT3 is added in Table 2 (page 8) and on page 12.
  • It is well known that NF-kB plays important roles in the regulation of cancer development, however, the roles of NF-kB controlled by miRNA/AKT3 axis in human cancer have not yet been well elucidated.
  • As reviewer suggested, a paragraph related to innovative and future pharmacological perspectives of using miRNAs is added on page 14 and page15.
  • A figure (Figure 3) that generally depicting the biochemical pathways involved in the cancer development modulated by miRNA/AKT3 is added on page 14.
  • Typographical errors are corrected by the editors of Science Journal (see attached certificate).

References:

1          Wu SJ, Chen J, Wu B, Wang YJ, Guo KY. MicroRNA-150 enhances radiosensitivity by inhibiting the AKT pathway in NK/T cell lymphoma. J Exp Clin Cancer Res 2018; 37: 18. doi: 10.1186/s13046-017-0639-5

2          Wang Z, Ma K, Pitts S, Cheng Y, Liu X, Ke X et al. Novel circular RNA circNF1 acts as a molecular sponge, promoting gastric cancer by absorbing miR-16. Endocr Relat Cancer 2019; 26: 265-277. doi: 10.1530/erc-18-0478

3          Wang X, Li GH. MicroRNA-16 functions as a tumor-suppressor gene in oral squamous cell carcinoma by targeting AKT3 and BCL2L2. J Cell Physiol 2018; 233: 9447-9457. doi: 10.1002/jcp.26833

Round 2

Reviewer 2 Report

Comments and Suggestions for Authors

Accepted